

# An innovative artificial neural network model for smart crop prediction using sensory network based soil data

Shabana Ramzan[1], Basharat Ali[2], Ali Raza[3], Ibrar Hussain[3,4], Norma Latif Fitriyani[5], Yeonghyeon Gu[5] and Muhammad Syafrudin[5]

[1] Government Sadiq College Women University Bahawalpur, Bahawalpur, Pakistan
[2] Agronomic Research Station Bahawalpur, Bahawalpur, Pakistan
[3] Department of Software Engineering, University of Lahore, Lahore, Pakistan
[4] Faculty of Engineering & Information Technology, Shinawatra University, Bangtoey Samkhok, Pathum Thani, Bangtoey, Thailand
[5] Department of Artificial Intelligence and Data Science, Sejong University, Seoul, Gwangjin-gu, Republic of Korea

Corresponding authors
Yeonghyeon Gu, yhgu@sejong.ac.kr
Muhammad Syafrudin,
udin@sejong.ac.kr

## ABSTRACT

A thriving agricultural system is the cornerstone of an expanding economy of agricultural countries. Farmers' crop productivity is significantly reduced when they choose the crop without considering environmental factors and soil characteristics. Crop prediction enables farmers to select crops that maximize crop yield and earnings. Accurate crop prediction is mainly concerned with agricultural research, which plays a major role in selecting accurate crops based on environmental factors and soil characteristics. Recently, recommender systems (RS) have gained much attention and are being utilized in various fields such as e-commerce, music, health, text, movies etc. Machine learning techniques can help predict the crop accurately. We proposed an innovative artificial neural network (ANN) based crop prediction system (CPS) to address the farmer's issue. The parameters considered during sensor-based soil data collection for this study are nitrogen, phosphorus, potassium, temperature, humidity, pH, rainfall, electrical conductivity, and soil texture. Python programming language is used to design and validate the proposed system. The accuracy and reliability of the proposed CPS are assessed by using accuracy, precision, recall, and F1-score. We also optimized the proposed CPS by performing a hyperparameter Optimization analysis of applied learning methods. The proposed CPS model accuracy for both real-time collected and state-of-the-art datasets is 99%. The experimental results show that our proposed solution assists farmers in selecting the accurate crop and producing at their best, increasing their profit.

## INTRODUCTION

The demand for food items is increasing as the world's population is growing. To maintain a sustainable equilibrium, agricultural productivity must be increased. In addition to providing the majority of essential meals, agriculture also provides a source of income.

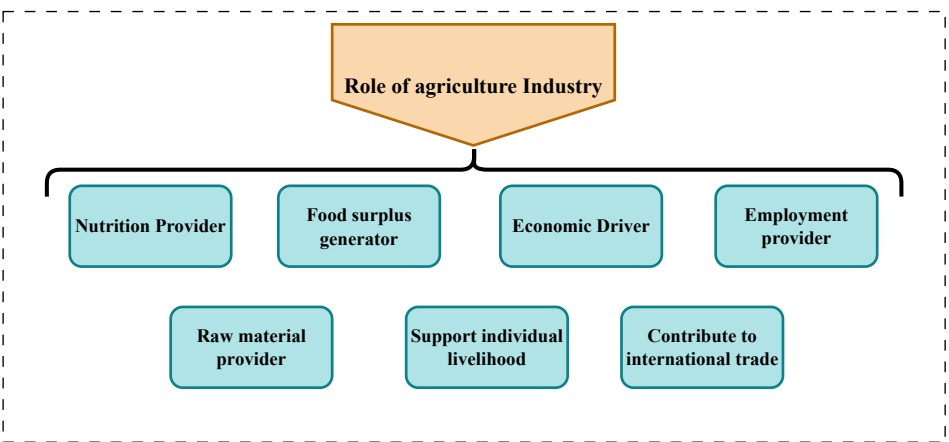

**Figure 1** Role of agriculture industry.

Farmers are suffering from poor yield and financial losses even in agricultural hubs due to incorrect selection of crops. These financial losses can be avoided by correct crop prediction, which considers environmental factors and soil characteristics. The country's agricultural industry is essential to its economic development as well as food security.

Agriculture uses natural resources to support human existence and generate revenue. Agricultural products are essential sources of the human diet and feed animals. Additionally, agriculture contributes to the world's economy by supplying essential items utilized in trade, such as dairy, cattle, grain, and raw resources for fuel. The agricultural industry has a very important role in human life and the development of the country, as shown in Fig. 1. The world's living things use agricultural waste and products as food (*Ferentinos, 2018*). When there is a change or trouble in the balance of agriculture, humanity, and the biological stock likewise face the biggest change or difficulty in maintaining the ecological cycle (*Dharani et al., 2021*).

The important elements for the expansion of agriculture include: There is a need to implement modern technologies to raise agricultural output. To maximize revenue and enhance agricultural productivity, farmers must acquire the necessary skills. Input characteristics related to soil and environmental conditions can have a great impact on crop output. For a correct crop selection, the soil's properties and the environment's state must be considered. Factors that form the soil have different impacts on soil formation as well as contribute to their value from an agricultural perspective (*Sawicka et al., 2017*), as shown in Fig. 1. To choose the best crop for land, the farmers need to be properly informed about all the variables, including soil pH, temperature, humidity, and other variables (*Chitragar et al., 2016*; *Warudkar & Dorle, 2016*; *Amrutha, Lekha & Sreedevi, 2016*; *Masrie et al., 2017*).

Farmers should also be aware of and take into account the elements that are essential for seed sowing as well as crop growth. Taking all pertinent factors into account will ensure the crop's health, a larger yield, and the highest possible income. Inadequate resources and poor farming decision-making are hurting the agriculture industry. There should be some system that assists all the stakeholders regarding farming operations. Primary

stakeholders and farmers lack the necessary knowledge to choose the best course of action for their farming operations. To select appropriate crops based on soil properties and climate variations, an automated decision-making system is needed. These methods will supplement the expertise and experiences of farmers (*Khatri-Chhetri et al., 2017*).

Earlier, farmers used their hands-on expertise to choose the crop, track its development, and when to harvest the crop. However, the agricultural community now finds it challenging to continue doing so in the modern day due to the quick changes in environmental circumstances such as global warming. As a result, there is a need to have automated decision-making methods that can supplant human methods for the correct selection of crops.

Accurate crop selection is one of the major challenges for farmers to increase their income as well as national income. Farmers should be able to make the crop selection according to different factors such as environmental factors, weather, and soil characteristics to get maximum yield. Unfortunately, farmers are not able to make confident decisions to select accurate crops due to a lack of knowledge. There is an essential need to develop an intelligent automated system that can make decisions on crop selection for farmers. IoT (Internet of Things)-based machine learning crop prediction can address the challenges of farmers for accurate crop selection to maximize their income.

## Research questions

The aforementioned discourse has presented the subsequent research problems and questions:

- How to develop a system for addressing the issues with conventional farming and do the selection of crops for the particular field by considering the environmental factors and soil characteristics?
- How to develop a user-friendly system that will enable farmers to comprehend efficient agricultural methods to overcome the limitations of traditional, regional approaches and meet the growing population's need for food?
- How to minimize the chance of crop failure and increase farmer's revenues?

When choosing what kinds of crops to plant and what to do throughout the crop-growing season, machine learning (ML) is a useful tool for decision-making. Numerous industries, such as finance, healthcare, agriculture, pharmaceuticals, personalized banking, and customer service, employ ML techniques and IoT. Taking into account these issues, we used IoT and ML algorithms, artificial neural network (ANN), to develop the CPS that would benefit farmers. For precision agriculture, an automated decision-making system is required that can be developed by using a powerful tool of computational intelligence (*Pierce & Nowak, 1999*; *Murase & Ushada, 2006*; *Van Alphen & Stoorvogel, 2000*).

## Research contributions and innovation

The following are our proposed approach's primary contributions:

- We design an optimized artificial intelligence (AI) system to predict the correct crop for a particular field based on different soil parameters.

- We design a real-time sensory network-based environment for data collection, which is based on soil parameters temperature, electrical conductivity (EC), water requirement and soil texture (dataset1), nitrogen, phosphorus, potassium, temperature, humidity, pH, rainfall (dataset 2) to overcome the limitations of traditional, regional approaches.
- An intelligent method based on ML, the ANN algorithm is used to choose the accurate crop for the field by considering environmental and soil characteristics.
- We use an optimized ANN approach to reduce the error rate during the classification process.
- To evaluate the performance of the proposed approach, different metrics are used, such as accuracy, precision, recall, and F1-score. State-of-the-art comparisons show our model outperformed.

The remaining manuscript has the following sections: Section 'Related Work' discusses the crop prediction systems and algorithms presented in the agricultural field. Section 'Proposed Methodology' details the architecture and components of the proposed approach. Section 'Results and Discussion' presents the findings and evaluation process to illustrate the system's performance. Section 'Conclusions and future work' presents the conclusion and future work.

## RELATED WORK

Advanced technology such as ML was deployed to develop the system for the farmers to provide them guidance for sowing the crops (*Kalimuthu, Vaishnavi & Kishore, 2020*). The data about the seed crops was collected. Parameters like temperature, moisture content, and humidity were considered which are important factors in achieving a reasonable growth of crops. The Android-based mobile application was developed that takes input from the user about parameters like temperature and automatically takes their location to start the prediction process.

The prediction of crop and yield by considering soil parameters that have a great impact on the production of crop production (*Ishak, Rahaman & Mahmud, 2021*). The proposed ML-based crop recommendation system is scalable and can be used for different crops (*Patil, Panpatil & Kokate, 2020*). Decision tree, naïve Bayes, and k-nearest neighbor (KNN) algorithm were used and KNN outperformed all three algorithms. The ML algorithms were used to develop the system to select the best crop for sowing by considering environmental and soil parameters (*Nischitha et al., 2020*). The system also guides the fertilizers and seeds for cultivation. It helps the farmers to use a new crop variety, may increase their profit, and also avoid soil pollution. The farmers are facing the major problem of proper crop prediction which can badly affect the crop productivity and their income.

ML algorithm such as artificial neural networks and support vector machines is used to develop the crop prediction system by considering the environmental and soil factors (*Fegade & Pawar, 2020*). The interface is designed to take input from farmers and predict the crop. The prediction accuracy of the Neural network is 86.80%. The farmers are currently dealing with two major challenges such as climatic changes and soil nutrient deficiency that are affecting crop growth. To get accurate crop prediction, it is necessary to

have proper information on agrometeorological parameters. Particular issues might arise from aspects of these factors' variability (*Marenych et al., 2014*). This problem has been addressed by several researchers, with different levels of success (*Grabowska et al., 2016*; *Li et al., 2021*).

The proposed research uses linear regression methods to increase the production and farmer's profit (*Yamparla et al., 2022*). The farmers grow the same crop year after year without going for other new varieties and use fertilizers without understanding their needs and quality. The system is designed for crop prediction, which can help farmers select the crop based on climatic conditions and soil nutrients (*Rao et al., 2022*). The study also compared different algorithms such as random forest classifier, decision tree, and KNN by using two different criteria entropy and GINI. The research findings show that random forest outperforms among the three. Climate change has a direct impact on crop yield and growth. The Fuzzy logic-based system was developed for cop yield prediction by considering climatic change parameters (*Borse & Agnihotri, 2019*). Humidity, evaporation, rainfall, and temperature parameters are considered for the prediction. The 15-year climatic variables and crop yield data are considered. The fuzzy model used a triangular membership function. This fuzzy rule-based system (FRBS) follows the approach called Takagi Sugeno-Kang to develop the model. The model is evaluated by a coefficient of correlation that is more than 0.9 between the actual and predicted yield.

The study proposed a novel approach using IoT techniques and the Crop Yield Prediction Algorithm (CYPA), in precision agriculture (*Talaat, 2023*). Crop yield simulations make it easier to understand the combined effects of field factors like nutrient and water shortages, pests, and diseases during the growing season. The proposed algorithm includes weather, climate, agricultural productivity, and chemical data to fulfill the expectations of farmers and policymakers about crop yields. The study performed the hyper-parameter tuning to have the best values for each ML method for model training and validation. ExtraTreeRegressor showed the best score of 0.9933. The CYPA's performance is enhanced by using a new algorithm that is based on active learning which reduces the labeled data number, required for training. This new algorithm improved the accuracy and efficiency of the system. The precision agriculture system used low input but achieved high accuracy using the Internet of things and machine learning for sustainable agriculture (*Parween et al., 2021*). The proposed system predicted the fertilizer accurately by different classifiers with corresponding heatmaps. Naive Bayes showed high accuracy because of dependency on probabilistic features. For better crop prediction, this classifier can be used.

The study focused on paddy crop nutrient deficiency. A neural network is built using the TensorFlow library to classify them into potassium, phosphorous, nitrogen deficiencies, or healthy independently (*Shidnal, Latte & Kapoor, 2021*). The optimal balance between potassium, phosphorous, and nitrogen is necessary. A set of images used by this TensorFlow model to identify the deficiencies. The deficiency level on a quantitative basis is estimated by feeding the result into ML driven layer. It mainly uses the k-means-clustering ML algorithm. The approach is evaluated through the rule matrix.

Crop prediction is the critical decision of farming. However, the agricultural community finds it challenging to do so these days due to the continuous changes in environmental

factors. As a result, ML approaches have supplanted prediction in recent years, and this work has employed several of them to predict crop production. To guarantee that a particular ML model operates with a high degree of accuracy, effective feature selection techniques are used to transform the raw data into a dataset that can be computed by ML easily (*Raja et al., 2022*). To design an accurate ML model, only those data features are selected that are most relevant in determining the output. The additional irrelevant features make the model complex and also increase its space and time complexity. The results showed that an ensemble technique provided higher accuracy than other classification models. The study compares several wrapper feature selection strategies to predict crop by using classification approaches that recommend the best crop (*Suruliandi, Mariammal & Raja, 2021*). The Recursive Feature Elimination method performs better than the others when combined with the adaptive Bagging classifier. The crop prediction system is designed using a sensor-based dataset and publicly available dataset with ML and ensemble learning (*Ramzan et al., 2024*).

The DL and ML-based proposed study provides the solutions to address the challenges of cultivation. It recommends the crops by considering the parameters of weather and also guides about the requirement of nutrients. Also, recommend the herbicides after identifying the weeds. Used publicly available datasets to develop four modules to recommend crops, identify weeds, recommend pesticides, and crop cost estimation. The proposed modules are implemented by using ML and DL algorithms that will help the farmer select the best-suited crop for their land (*Durai & Shamili, 2022*). The proposed research represented the importance of modern IoT-based ML systems for agriculture. The results presented that real-time data and ML algorithms help farmers make decisions about the factors that have a strong impact on crop growth. Investigated the crops using ML algorithms by considering general characteristics. The research findings indicate that accurate feature selection of agriculture data improves the accuracy of ML algorithms. The proposed system provides help to get more crop yield (*Elbasi et al., 2023*).

The proposed study fine-tuned the random forest algorithm and also integrated long short-term memory interconnected networks to get an accurate prediction of the crop. Used localized weather forecasts with different analysis techniques to bring breakthroughs in smart agriculture practices. This approach helps the farmer to utilize the resources in a better way to manage the crop and agriculture product's yield (*Mahale et al., 2024*). The use of AI and IoT to improve crop prediction and yield. Different sensors are implemented to collect real-time data about environmental factors and soil characteristics of the particular agricultural field to train the model. The communication network is generated to transfer data from sensors to the Al model that is already trained to perform analysis. The system helps the farmers to get more yield by making appropriate crop selection decisions and detecting crop disease timely (*Priya et al., 2024*).

Machine learning-based crop prediction systems help farmers to reduce the effective cost, increase their profits, and control crop failure (*Elbasi et al., 2023*). Automated crop prediction systems by considering the climate/weather/environmental factors and soil parameters play a crucial role in minimizing crop losses, disasters between 2008 and 2018 became the major cause of crop and livestock decline of USD 108.5 billion in

lower income and also lower-middle-income countries (LMICs), the impact of climate changes can further increase it (*Canton et al., 2021*; *Markhof, Ponzini & Wollburg, 2022*). Environmental data-based crop prediction systems increase the crop yield by selecting the suitable crop accordingly, in 2010,2012 U.S. Corn Belt yield affected by high temperatures in nighttime and warm winter became the cause of losses of 220 million dollars of Michigan cherries in 2012 (*EPA, 2017*). ML and DL-based automated crop prediction systems make the automatic crop selection to reduce output losses of agriculture productions (*Sharma et al., 2023*).

# PROPOSED METHODOLOGY

The proposed system's architecture illustrates how each of its components interacts with the others, as shown in Fig. 2. The proposed system is based on two datasets: one is hybrid data collection (real-time data from sensors plus manual data), and the other is a publicly available dataset, containing 1101 data points. The real-time sensing of the environmental and soil parameters using sensors for crop prediction. The ML algorithm, ANN, is used for analysis and prediction.

## Architecture of CPS

The proposed system comprises two major functional components data input and analysis. It allows us to take crop prediction factors in real-time input as a hybrid dataset (from sensors and manually, Dataset 1) and static dataset (publicly available, Dataset 2) according to the working rules devised with the expert's opinion, given in Table 1. For CPS, real-time sensing is the responsibility of the IoT component, while data analysis is the responsibility of the ML algorithm, ANN, this component allows to make crop predictions. Dataset 1 and Dataset 2 are used as input for ANN. The trained ANN takes the input and does a comparison with the dataset column(dependent) value to classify them to which crop class it belongs. The system uses working rules for Dataset 2 and the label column of Dataset 1.

## Real time data collection hardware design

The proposed approach hardware is based on an Arduino-based circuit with sensors to measure environmental and soil features for crop prediction. The sensors are used to sense real-time data for air temperature and soil EC. The Arduino platform is used for hardware design, we used the DHT 22 temperature and humidity sensor for air temperature and the Mec10 soil EC sensor used for soil. A suitable type of soil is required for every crop for sowing and growing. Therefore, we collect data from different regions according to the suitability of soil for crops. Figure 3 shows the circuit diagram.

### DHT 22 temperature and humidity sensor

We used a DHT-22 sensor to measure air temperature, a capacitive type temperature sensor with high precision and low cost. It is a digital–output sensor, and as compared to DHT-11, over the large range, it has a more accurate sensing capacity. Its technical characteristics are given in Table 2.

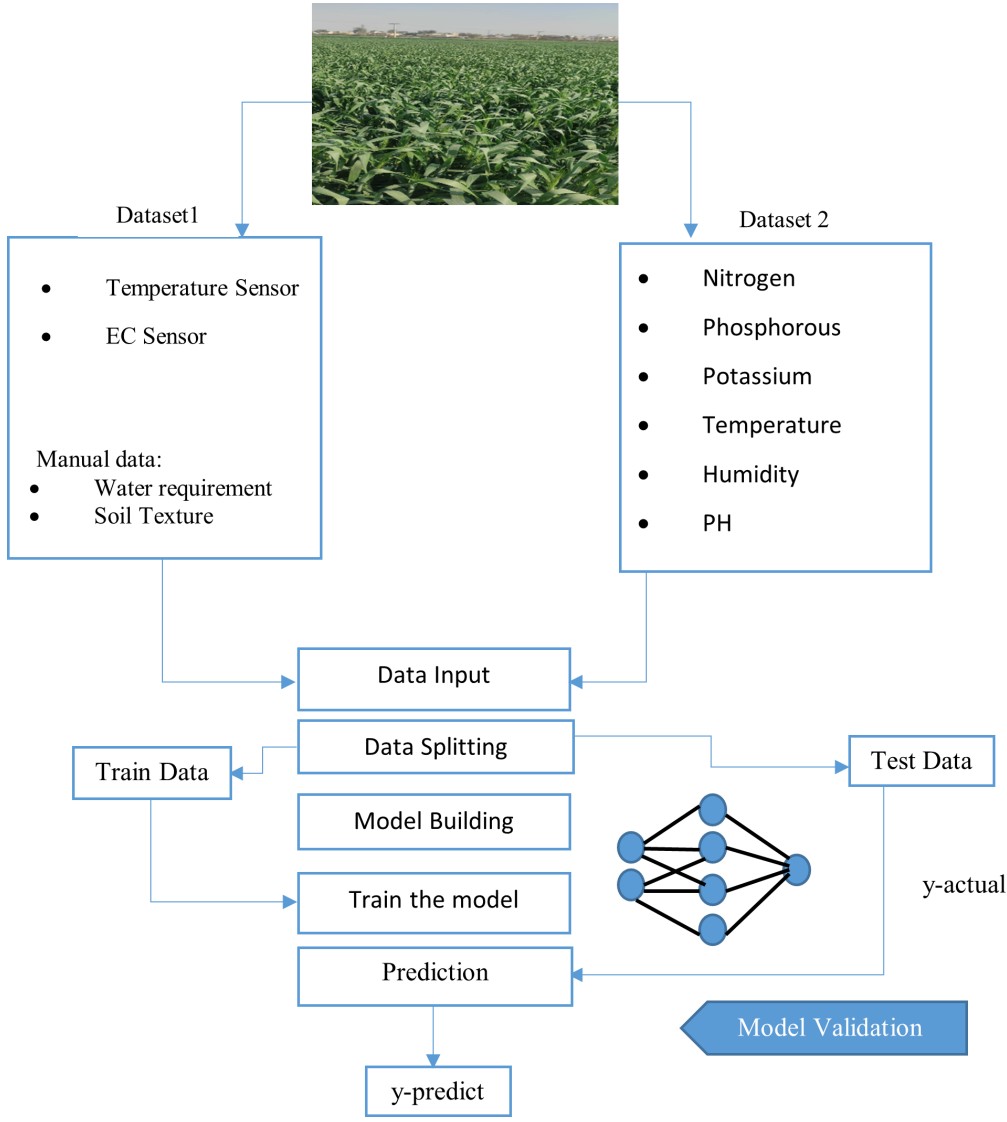

**Figure 2** Architecture design of CPS.

### Mec10 Soil EC sensor

We used the Mec10 sensor to measure soil EC because it is a stable and reliable sensor for the measurement of soil EC, as shown in Fig. 4. Its technical characteristics are given in Table 3.

## Dataset features description

The Dataset 1 features description is given below:

- Temperature: To determine the evaporation rate, the temperature is an important factor. For example, the evaporation rate is high when the temperature is high in the daytime. When evaporation rate is high more water is required. At night when the

| Table 1 | Working rules for collected Dataset 1. | | | | |
|---|---|---|---|---|---|
| Sr# | Crops | Temperature (Co) | Water requirement (mm) | Soil texture | Electrical conductivity (EC) |
| 1 | Wheat | 15–28 | 300 | All soil except heavy soil | 4 |
| 2 | Rice | 30–40 | 1200 | Clay, silty clay, clay loam | 4 |
| 3 | Cotton | 30–40 | 500–800 | Loam soil, medium heavy | 4 |
| 4 | Maize | 20–38 | 500–800 | silty loam, loamy, medium | 4 |
| 5 | Gram | 20–30 | 200 | silty loam,sandy loam | 4 |
| 6 | Groundnut | 25–35 | 600 | Loam, medium clay | 4 |

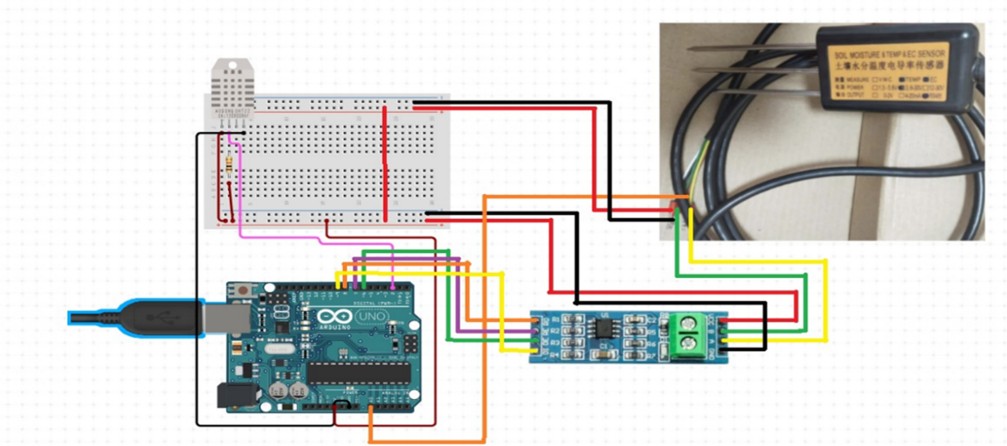

**Figure 3** The circuit diagram.

| Table 2 | Technical characteristics of DHT-22 sensor. |
|---|---|
| **DHT 22 temperature and humidity sensor** | |
| Temperature | 40 to 80 ° C |
| Accuracy | ±0.5 ° C |
| Power | 3–5 V |

evaporation rate is low, less water is required as compared to day time. This is the reason temperature is considered an important factor in predicting the crop.

- EC: It is significant for crop plants for several reasons, the most important one is the plant's capacity to absorb nutrients and water from the water solution or soil. Therefore, it is important for crop prediction.

- Soil texture: It affects nutrient absorption and has a significant role in the management of nutrients. For example, soils with finer textures typically have higher soil nutrient storage ability. This is the reason soil texture is considered an important factor in predicting the crop.

The Dataset 2 features description is given below:

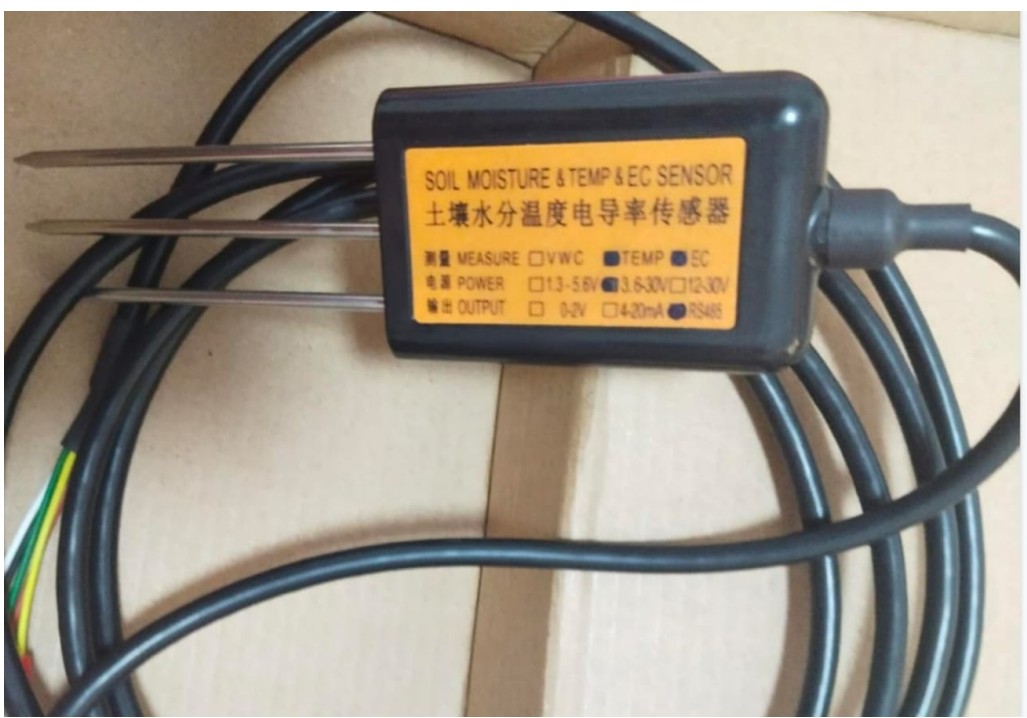

**Figure 4** Soil moisture, temperature, and EC sensor.

**Table 3** Technical characteristics of Mec-10 sensor.

| MEC10 Soil EC sensor | |
| --- | --- |
| EC | 0-5000 us/cm,10000us/cm, 20000us/cm |
| Accuracy | 0-10000, ±3% ; 10000-20000us/cm, ±5% |

- Temperature: To determine the evaporation rate, the temperature is an important factor. For example, the evaporation rate is high when the temperature is high in the daytime. When evaporation rate is high more water is required. At night when the evaporation rate is low, less water is required as compared to day time. This is the reason temperature is considered an important factor in predicting the crop.
- Nitrogen (N): It is extremely important for crops and accordingly for crop prediction because it makes up a large portion of chlorophyll for crop plants, which is the substance that allows plants to use solar energy to produce sugar.
- Phosphorus (P): It is one of the most important minerals for crop prediction, a component of plant cells that is necessary for cell division and the health of the plant's growing tip. It is essential for seedlings and young plants.
- Potassium (K): It is very important to control the exchange of oxygen, carbon dioxide, and water vapor. Plant development is stunted and yield is decreased if K is insufficient or not provided in sufficient proportions. This is the reason it is considered an important factor in predicting the crop.

- Humidity: It reflects the environmental moisture content and plays an important role in predicting the crop.
- pH: Soil pH is important to predict the correct crop for sowing in a particular field.
- Rainfall: According to FAO (Food and Agriculture Organization of the United Nations.) guidelines, regions with less than 450 mm of annual rainfall are not suitable for agriculture due to the high water needs for plant development in the absence of irrigation. It is a very important source of water for crops and has a vital role in predicting the accurate crop.

## Exploratory data analysis

Heatmaps are generally used to identify the best feature within a dataset to build a prediction model. The variable density and intensity are displayed in the correlation matrix. As Fig. 5. Illustrate the correlation for Dataset 1; the low numbers are shown in light color, while the high values are displayed in dark blue color. As Fig. 6 illustrates, the correlation for Dataset 2, there is a strong correlation between the potassium and phosphorus levels, the low numbers are shown in light yellow, while the high values are displayed in dark orange.

## Data preprocessing

To prepare the data for decision-making, data preprocessing is performed. We have examined our datasets to make sure that there are no null or missing values. There are several data scaling methods, we have used standardization that rescaling the distribution of values to make the standard deviation equal to one and the mean of the observed values equal to zero, given by Eq. (1).

$$A_{\mathrm{MMS}} = \frac{A - A_{\mathrm{mean}}}{\sigma} \tag{1}$$

where:

$A_{MMS}$: is the new/ resulting value of the feature.
$A$: The original/actual value of the feature.
$A_{\mathrm{mean}}$: is the mean of the feature/column.
$\sigma$: is the standard deviation of the feature/column

We apply the label encoder to convert classes into numeric numbers for the crop column in Dataset 1(0-5) and the label column in Dataset 2(0-21).

## Data splitting

The dataset splits into two subsets, training, and test set, by using a random split function. The training set is used for the model training, and the test set is used for the validation and evaluation of the model. These two sets split using a ratio of 0.2, which means 80% training dataset and 20% test dataset.

## Applied artificial neural networks based CPS

The proposed system designed and trained the ANN model. The training dataset is used to train the model for crop prediction. In addition, we have built several state-of-the-art machine-learning methods such as logistic regression (LR), AdaBoost (AB), Gaussian naive Bayes (GNB), and support vector classifier (SVC).

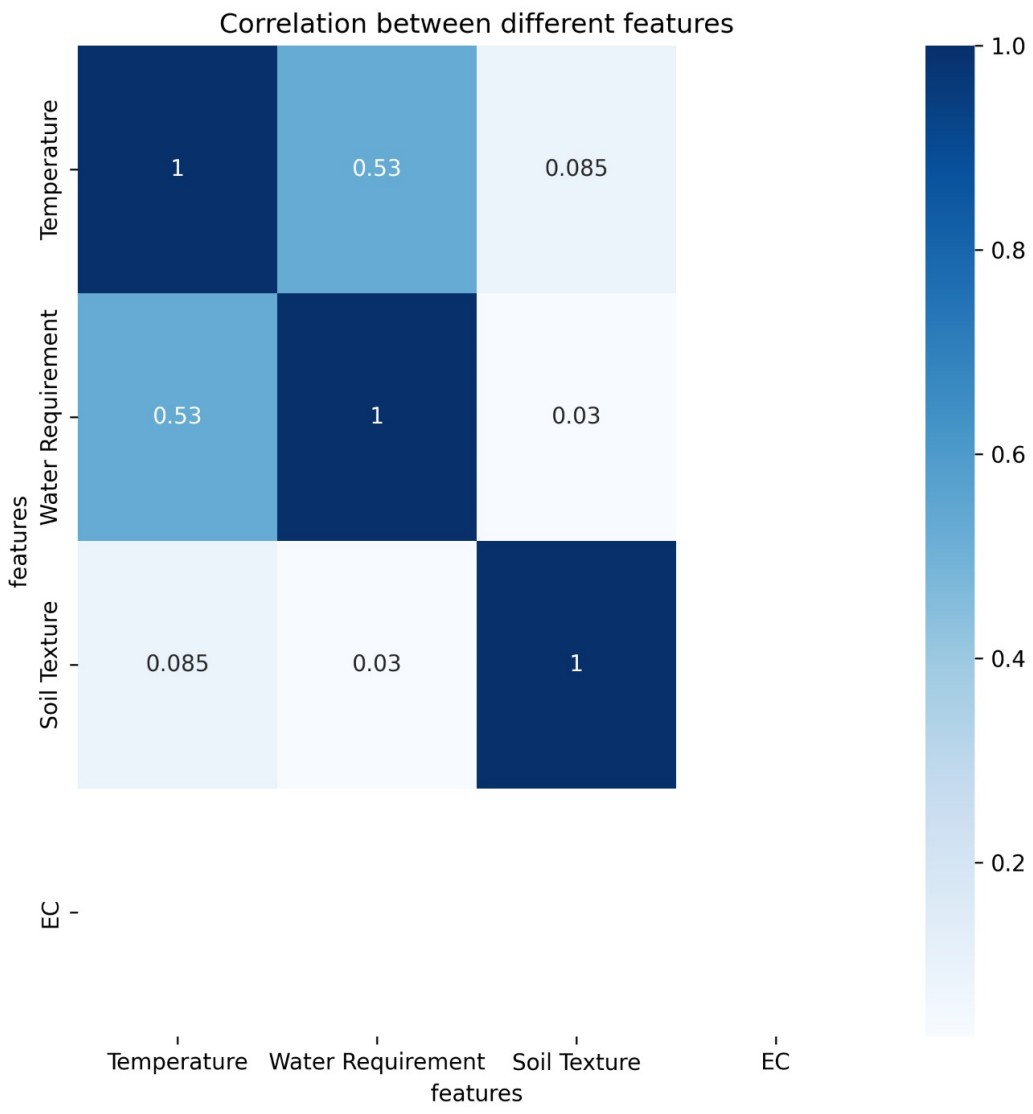

**Figure 5** Correlation matrix for Dataset 1.

### Proposed optimized artificial neural network

The structure of biological neural networks, in which the neuron is the fundamental brain processing unit, is the basis of artificial neural networks (*Akhtar et al., 2024*; *Raza et al., 2024*). Artificial neural networks also have neurons that are connected in different layers of the networks, much like neurons are in the human brain. The input, hidden, and output layers are the three layers of an ANN (*Zheng et al., 2019*; *Ahmadi et al., 2022*; *Goceri, 2021*). After receiving an input, the input layer passes it on to the hidden layer, which produces the result after executing the selected activation function and sends it to the output layer. The hidden layer performs various transformations on the input to produce output. The ANN configuration depends upon the problem's nature and the required output and problem

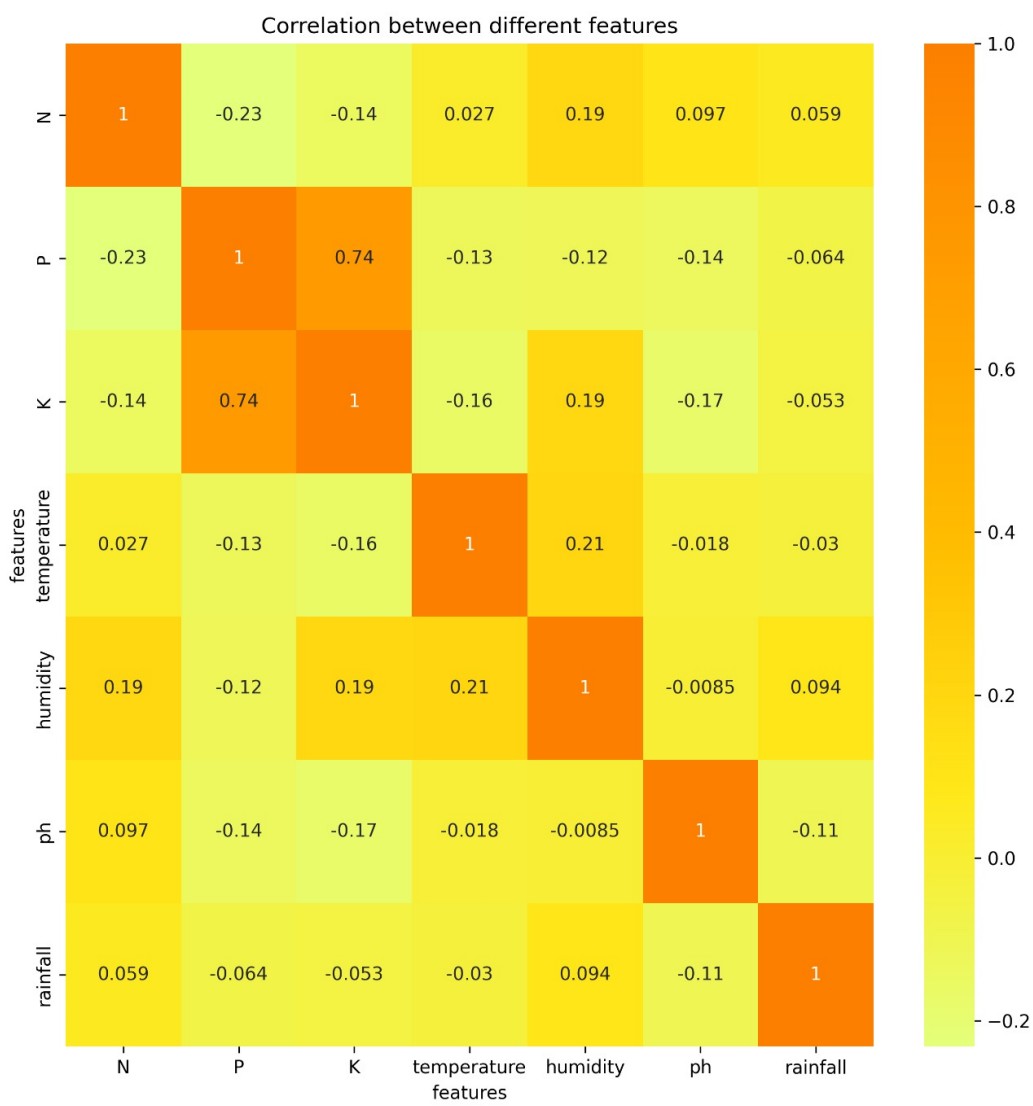

**Figure 6** Correlation matrix for Dataset 2.

(*Zhong et al., 2019*).

$$S = \sum_{i=1}^{n} W_i X_i + \text{Bias}$$

(2)

where:

S = Summation symbol.

$W_i$ = Weights of input variables

$X_i$ = Input variables

Bias = $W_0$ or intercept

The weighted total is determined and sent as an input to an activation function, which generates the output. Summation and activation function performed by the neuron.

Activation functions decide which nodes should fire and make it to the output layer. Depending on the nature of the problem, different activation functions can be applied.

### Proposed model building algorithm

The compile function is used to build an ANN model, the model is trained on a training dataset, and the model is validated using a test dataset. The model predicts the correct crop according to the independent features of the dataset. Python 3.11 is used to implement the ANN model, and the following two Python libraries, TensorFlow 2.13.0 and Keras 2.13.0, are used. The Python working algorithm is given as Algorithm 1.

---

**Algorithm 1** Python algorithm for ANN implementation

---

1: Take the input
2: Define dependent (target) and independent (predictor) variables
3: Define the input layer as the first hidden layer
4: Define the second hidden layer
5: Define output layer
6: Build the model using the compile function
7: Split the input dataset into training and test subsets
8: Train the model on the training subset
9: Validate the model using the test subset

---

According to the algorithm, we load the dataset as an input dataset for the ANN model. The input dataset 1 includes four features such as temperature, water requirement, EC, and soil texture as predictor variables and crop as output/target variable, which is encoded as 0–5, for dataset 2 include seven features such as N, P, K, temperature, humidity, ph and rainfall as predictor variables and label as output/target variable which is encoded as 0–21. The dataset is in Excel/ CSV file, after loading the dataset, the ANN model was then created by adding three layers to this model and using the classifier compile () method. Input is split into two subsets: training subset for model training and attest subset for validating the model once it has been trained.

### Models hyperparameters optimization

We have optimized the applied neural network techniques using a GridSearch CV approach. Furthermore, this optimization was achieved through recursive processing of training and testing data. The optimized ANN is utilized for crop prediction, enhancing the accuracy and reliability of the forecasting model. The best-fit hyperparameters are described in Table 4.

## RESULTS AND DISCUSSION

In this results evaluation section, the results for an ANN-based crop prediction model are discussed. It is crucial to assess the model's accuracy and reliability. Here, we detail the

**Table 4 Hyperparameter optimization analysis of applied learning methods.**

| Technique | Hyperparameter description |
|---|---|
| LR | random_state=10, solver='lbfgs',max_iter=6, multi_class='auto', $C = 1.0$ |
| AB | $n_e stimators$=50,$learning_r ate = 1.0, algorithm = SAMME.R$ |
| GNB | var_smoothing=3 |
| SVC | kernel='rbf',degree=3, gamma='scale', decision_function_shape='ovr' |
| Optimized ANN | activation="softmax", loss="sparse$_c$ategorical$_c$rossentropy", optimizer='adam', metrics=['accuracy'], $batch_s ize = 25$, $epochs = 50$ |

performance of our ANN and other applied models through various statistical measures and comparisons with baseline models.

## Experimental setup

To evaluate the effectiveness of our proposed approach, we have calculated the accuracy, recall, precision, and F1-score.

### Accuracy

The model's accuracy is measured by dividing the true predictions by the total predictions, as given in Eq. (3).

$$Accuracy = \frac{T_p}{T_p + T_N + F_p + F_n} \tag{3}$$

where Tp is the true positive prediction, Tn is the true negative prediction, Fp is the false positive prediction, and Fn is the false negative prediction.

### Recall

To measure the recall (sensitivity), the true positive predictions are divided by the sum of true positive predictions and false negative predictions, as given in Eq. (4).

$$Recall(R) = \frac{T_p}{T_p + F_n}. \tag{4}$$

### Precision

To calculate the precision, the true predictions are divided by the sum of all true positive and true negative predictions, as given in Eq. (5).

$$Precision(P) = \frac{T_p}{T_p + F_p}. \tag{5}$$

### F1-score

To calculate the F1-score, use precision and recall, the formula is given in Eq. (6).

$$F_1\text{-score} = 2 \times \frac{P \times R}{P + R}. \tag{6}$$

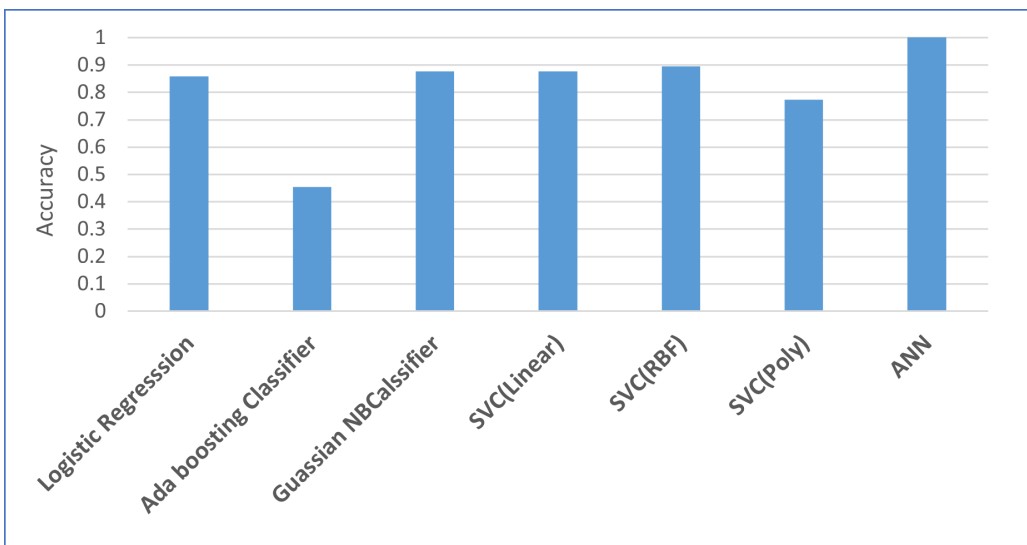

**Figure 7** Accuracy comparisons of ANN with other ML algorithms for Dataset 1.

## Results of applied baseline methods

The results of applying baseline classical methods for both datasets for crop prediction are analyzed in this section. We used several methods, including logistic regression (LR), AdaBoost, Gaussian naive Bayes, and SVM. The chart-based analysis in Figs. 7 and 8 shows the results of the applied baseline methods compared to our proposed artificial neural network (ANN) approach. The analysis demonstrates that our proposed approach outperformed the state-of-the-art classical methods for both datasets. For dataset 2, most of the algorithm shows more than 90% accuracy but ANN shows higher accuracy among all, 99%.

## Training results of ANN

We evaluated the proposed ANN model for prediction on our training dataset. To see the training and validation accuracy and loss of the ANN model, it is trained using various epochs and batch sizes. The training and validation loss is shown in Fig. 9A for Dataset 1 and Fig. 9B for Dataset 2; loss and epoch size have an inverse relationship, meaning that loss may be reduced by increasing epoch size and vice versa. The training and validation accuracy is shown in Fig. 9C for Dataset 1 and Fig. 9D for Dataset 2, illustrating the direct relationship between accuracy and epoch size, meaning that an increase in epoch size leads to an improvement in accuracy and vice versa.

## Testing results of ANN

The testing results of the proposed ANN for the unseen test set are analyzed in this section. The main classification metrics are displayed for each class in the classification report. This provides a deeper understanding of the classifier's behavior than global accuracy, which might hide functional flaws in one class when dealing with multi-class problems. Tables 5 and 6 show the classification report, including precision, recall, and F1-score values of the

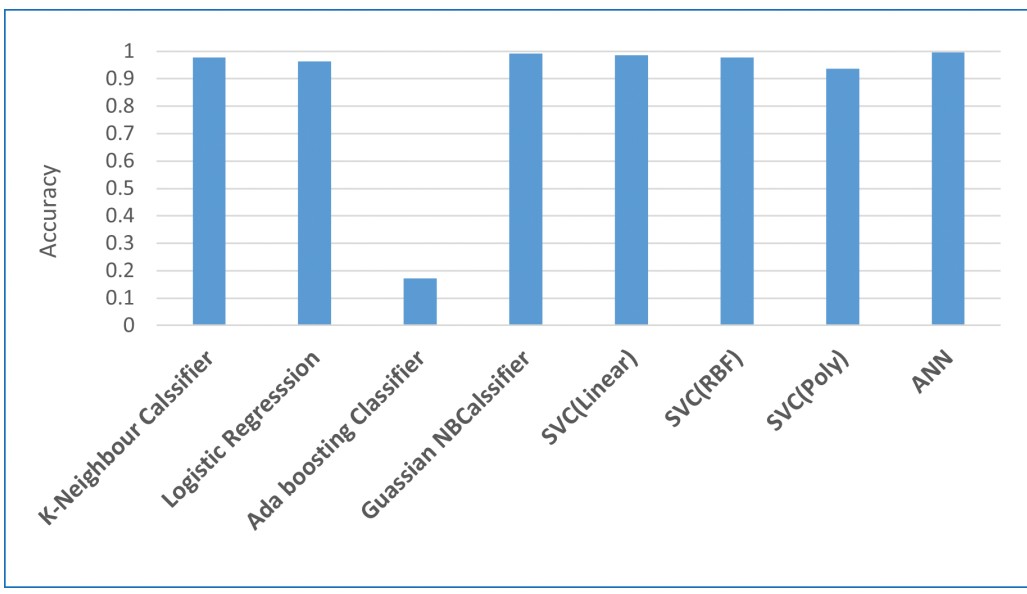

**Figure 8**  Accuracy comparisons of ANN with other ML algorithms for Dataset 2.

**Table 5**  Classification report for Dataset 1.

| Accuracy | Target class | Precision | Recall | F1 | Support score |
|---|---|---|---|---|---|
| | 0 | 1.00 | 0.97 | 0.99 | 39 |
| | 1 | 1.00 | 1.00 | 1.00 | 38 |
| | 2 | 1.00 | 1.00 | 1.00 | 40 |
| 0.99 | 3 | 1.00 | 1.00 | 1.00 | 56 |
| | 4 | 0.96 | 1.00 | 0.98 | 23 |
| | 5 | 1.00 | 1.00 | 1.00 | 24 |
| | **Average** | 0.99 | 1.00 | 0.99 | 220 |

ANN model for Dataset 1 and Dataset 2. The analysis concludes that the proposed ANN achieved high-performance scores for both datasets.

## Histogram based class wise performance analysis

The target class-wise performance results of the proposed CPS for both Dataset 1 and Dataset 2 are analyzed. Performance graphs of the CPS are shown in Figs. 10 and 11 for the classes of Dataset 1 and Dataset 2, respectively. We also conducted experiments to compare our findings with other ML models. Based on this comparison, we can conclude that the ANN model achieved better accuracy. For Dataset 2, most algorithms showed more than 90% accuracy, but the ANN model demonstrated the highest accuracy at 99.55%.

The dataset vise performance measures are demonstrated in Table 7. The results conclude that the accuracy result of the ANN model is improved for each data.

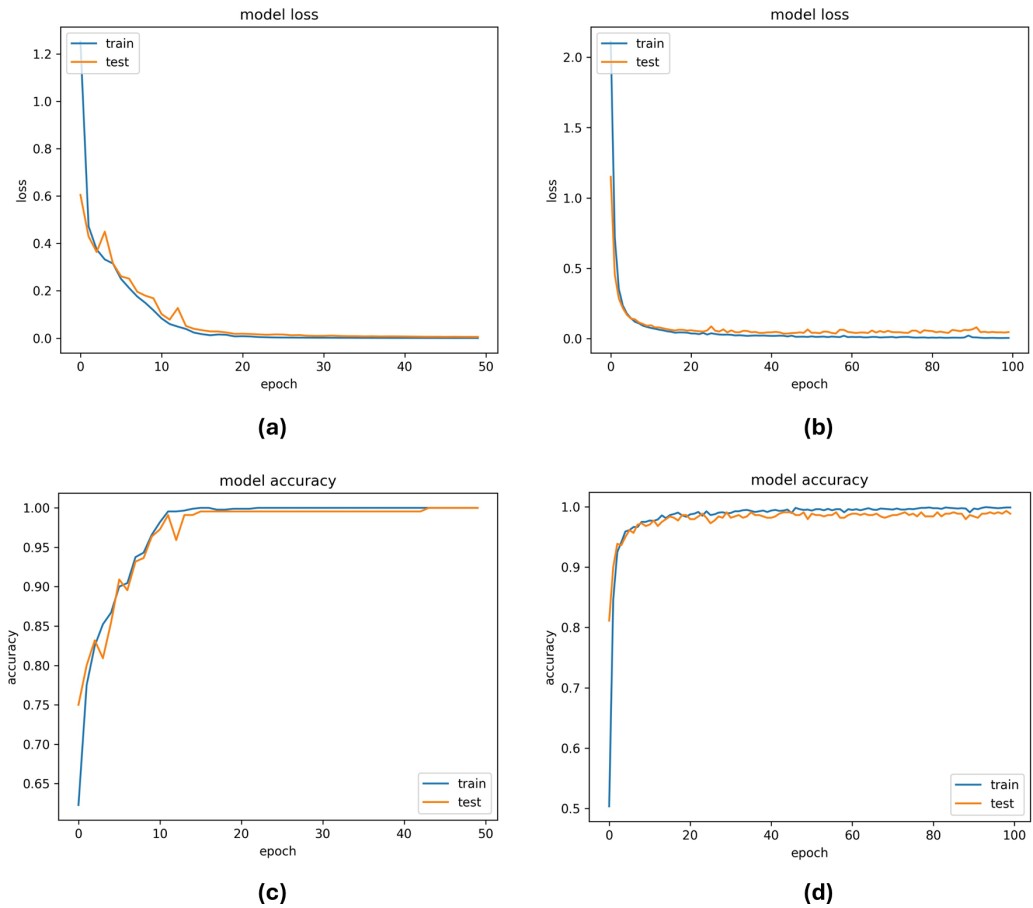

**Figure 9** The training and validation loss curves analysis (A) Loss curves for Dataset 1; (B) loss curves for Dataset 2; (C) accuracy curves for Dataset 1; (D) accuracy curves for Dataset 2.

## Confusion matrix validation

The confusion matrix-based performance validations are shown in Figs. 12 and 13. Performance accuracy can also be visually represented using a confusion matrix, which provides a visual interpretation of the proposed classifier's performance (*Deng et al., 2016*). The matrices for Dataset 1 and Dataset 2 are displayed, respectively. The diagonal values of the matrices indicate the correct predictions. The analysis shows that the proposed ANN achieves minimum error rates in terms of false classifications for corporate recommendations.

## State of the art comparison

We conducted a state-of-the-art research comparison analysis. A concise comparison of CPS with existing work is provided in Table 8. The minimum performance score achieved by precise methods is 86%. Our proposed approach outperformed the state-of-the-art methods, achieving high-performance scores of 99.5% for crop predictions.

**Table 6  Classification report for Dataset 2.**

| Accuracy | Target class | Precision | Recall | F1 |
|---|---|---|---|---|
|  | 0 | 1.00 | 1.00 | 1.00 |
|  | 1 | 1.00 | 1.00 | 1.00 |
|  | 2 | 1.00 | 1.00 | 1.00 |
|  | 3 | 1.00 | 1.00 | 1.00 |
|  | 4 | 0.94 | 1.00 | 0.97 |
|  | 5 | 1.00 | 1.00 | 1.00 |
|  | 7 | 1.00 | 1.00 | 1.00 |
|  | 8 | 0.91 | 0.95 | 0.93 |
|  | 9 | 0.95 | 1.00 | 0.98 |
|  | 10 | 0.94 | 1.00 | 0.97 |
| 0.99 | 11 | 1.00 | 1.00 | 1.00 |
|  | 12 | 1.00 | 1.00 | 1.00 |
|  | 13 | 1.00 | 0.96 | 0.98 |
|  | 14 | 1.00 | 1.00 | 1.00 |
|  | 15 | 1.00 | 1.00 | 1.00 |
|  | 16 | 1.00 | 1.00 | 1.00 |
|  | 17 | 1.00 | 1.00 | 1.00 |
|  | 18 | 1.00 | 0.95 | 0.98 |
|  | 19 | 1.00 | 1.00 | 1.00 |
|  | 20 | 1.00 | 0.92 | 0.96 |
|  | 21 | 1.00 | 1.00 | 1.00 |
|  | Average | 0.99 | 1.00 | 0.99 |

## Study limitations

Despite achieving the highest performance measures with the proposed approach, more soil parameters, such as nutrient retention capacity, can be considered. In addition, further deep neural networks can be built for more precise classification results.

## CONCLUSIONS AND FUTURE WORK

In the current study, we have used a machine learning algorithm, ANN, to predict the correct crop by considering the environmental factors and soil characteristics. The ANN-based system provided a more robust, effective, and reliable crop prediction system. The primary objectives of this study were to develop a user-friendly system that will enable farmers to comprehend efficient agricultural methods to overcome the limitations of traditional, regional approaches and meet the growing population's need for food, do the selection of crops for particular fields according to the environmental factors and soil characteristics and minimize the chance of crop failure and increase farmer's revenues. The proposed approach used environmental factors and soil characteristics based on data that has the following features: N, P, K, temperature, humidity, pH, rainfall, water requirement, EC, and soil texture. The proposed approach was evaluated using different evaluation metrics such as accuracy, precision, recall, and F1-score. To achieve better generalization by avoiding biases. The proposed approach will be helpful for farmers in making accurate selections

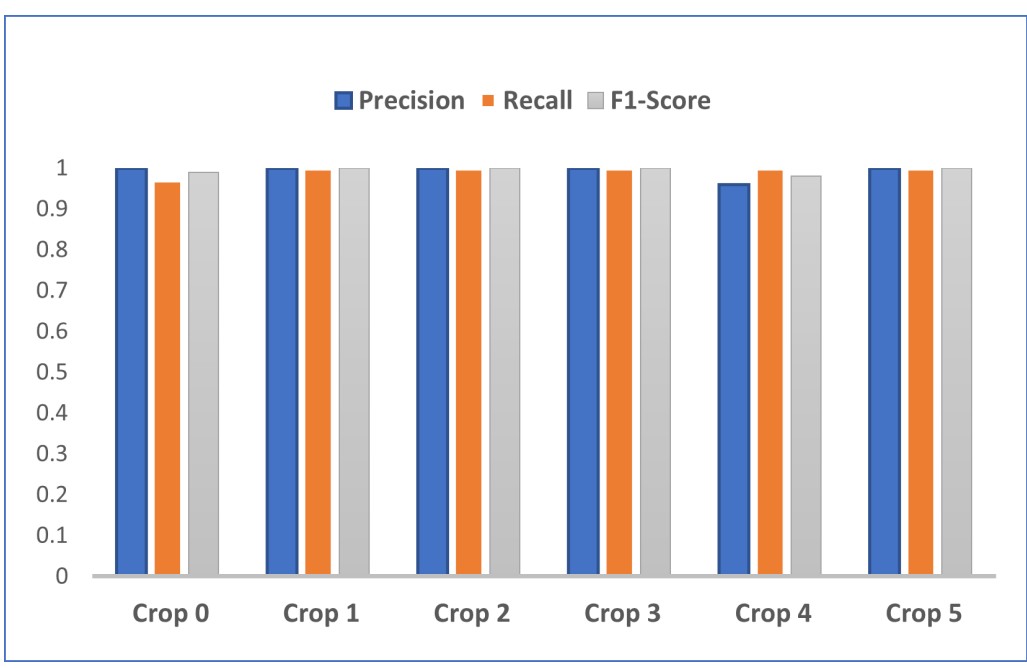

**Figure 10**  Performance graph for classes of Dataset 1 with CPS.

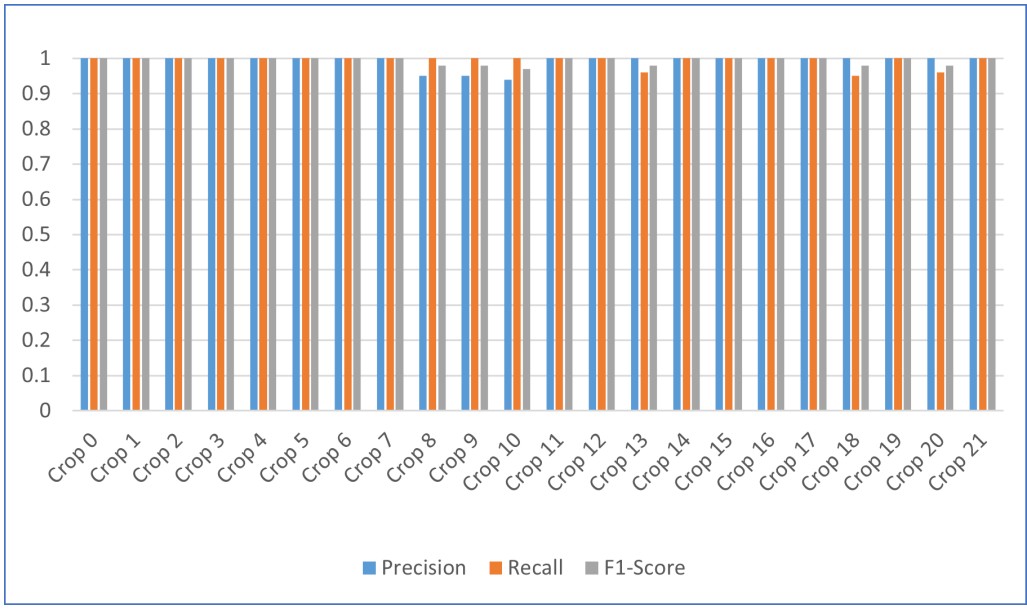

**Figure 11**  Performance graph for classes of Dataset 2 with CPS.

**Table 7   Dataset vise performance measures.**

| Model | Accuracy (dataset 1) | Accuracy (dataset 2) |
|---|---|---|
| Logistic Regresssion | 0.859 | 0.9772 |
| Ada boosting Classifier | 0.4545 | 0.9636 |
| Guassian NBCalssifier | 0.8772 | 0.1727 |
| SVC (Linear) | 0.8772 | 0.9931 |
| SVC (RBF) | 0.8954 | 0.9863 |
| SVC (Poly) | 0.7727 | 0.9772 |
| ANN | 99.99 | 0.9363 |

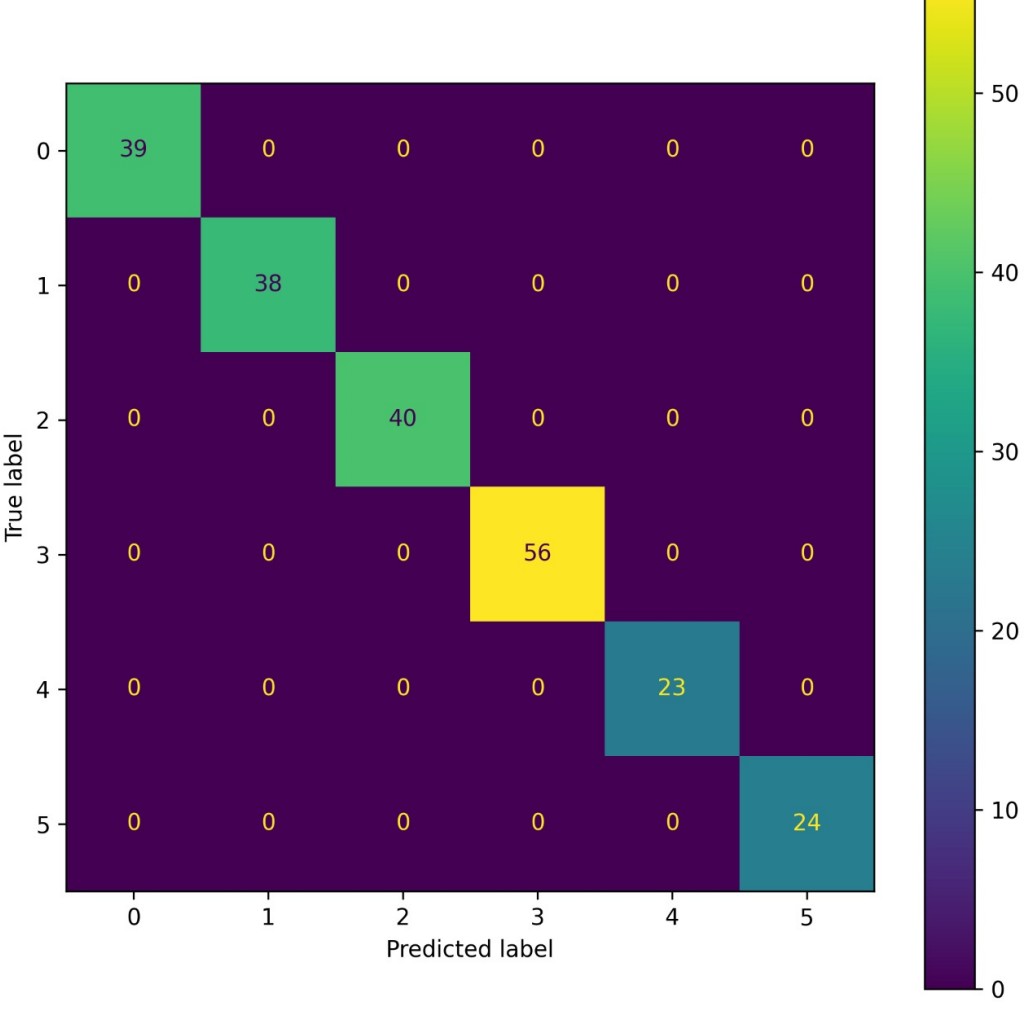

**Figure 12   Confusion matrix for Dataset 1.**

of crops for particular fields by considering environmental factors and soil characteristics, and it will also increase the yield of products and their profits.

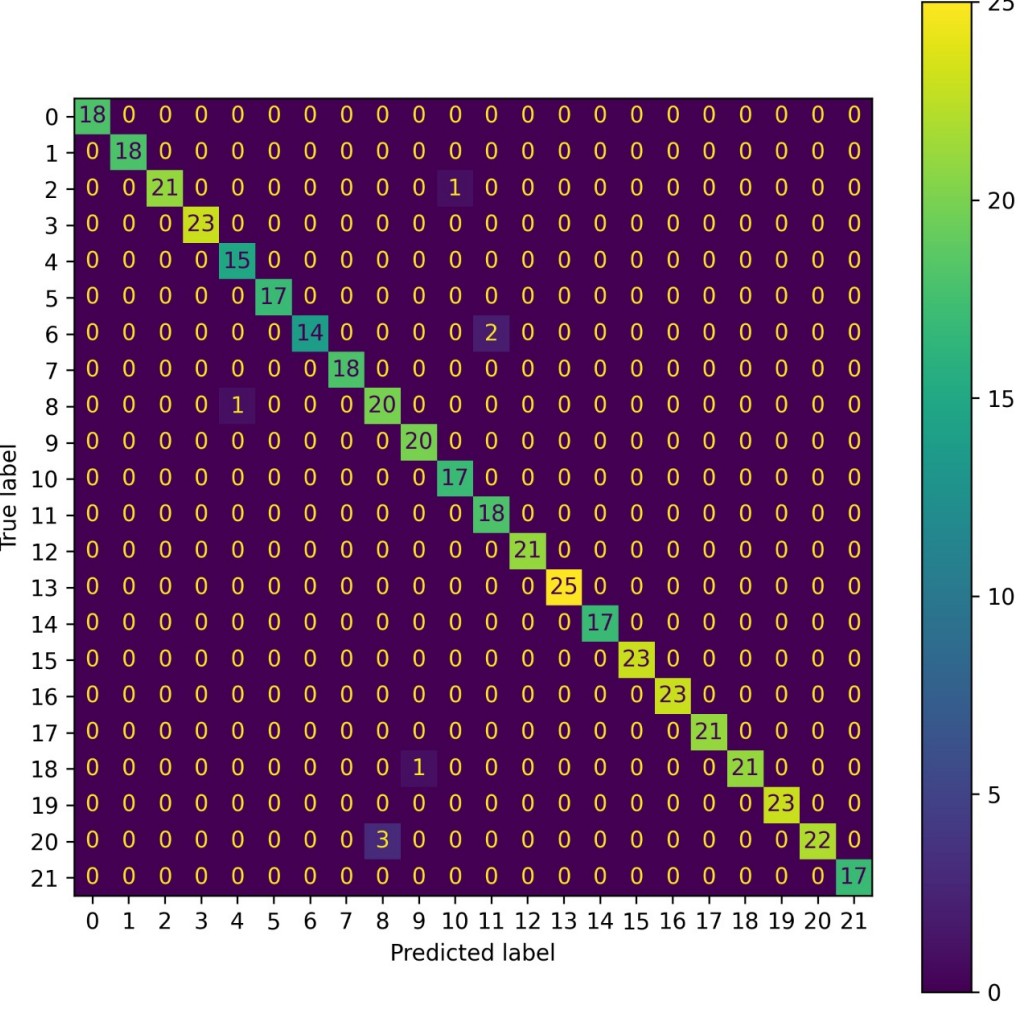

**Figure 13  Confusion matrix for Dataset 2.**

## Future work

For future work, we can use other analysis techniques, and they may perform better than artificial neural networks for decision-making. We can also consider other soil parameters such as nutrient retention capacity, heavy metal contamination, and availability of oxygen to roots.

Ramzan et al. (2024), *PeerJ Comput. Sci.*, DOI 10.7717/peerj-cs.2478

**Table 8** A brief comparison of CPS with existing works.

| Study | Study goal | Parameters | Technique | Performance | Limitations |
|---|---|---|---|---|---|
| *Suruliandi, Mariammal & Raja (2021)* | Crop prediction | Soil and environmental characteristics | Machine learning with recursive feature elimination method | Accuracy 97.29% | The humidity factor is not considered. |
| *Patil, Panpatil & Kokate (2020)* | Crop prediction | Climatic parameters and soil parameters like rainfall, temperature, moisture, and soil contents. | Machine learning | Accuracy 89.4% | Water requirement and EC are missing. |
| *Nischitha et al. (2020)* | Crop prediction | Soil ph, temperature, humidity, rainfall, crop data, and NPK values. | Machine learning | No performance metric used just prediction. | Soil texture, EC, and water requirements are missing. |
| *Fegade & Pawar (2020)* | Crop prediction | Rainfall, minimum and maximum temperature, soil type, humidity, and soil pH value | Machine learning and artificial neural networks | Accuracy 86.26% | Water requirement and EC is missing. |
| *Raja et al. (2022)* | Crop prediction | Soil and environmental characteristics | Machine learning | Accuracy 92.72% | The parameters considered are not mentioned. |
| *Parween et al. (2021)* | Crop prediction | Climate and soil factors. | Machine learning and IoT | Accuracy 96% | The humidity factor is not considered. |
| *Rao et al. (2022)* | Crop prediction | Climatic conditions and soil nutrients | Machine learning | Accuracy 98% | Soil texture, water requirements, and EC are missing. |
| *Ravichandran & Koteeshwari (2016)* | Agricultural crop predictor | Soil ph, NPK, depth temperature, and rainfall. | Artificial neural networks | Accuracy 92% | Soil texture, water requirement, humidity, and EC are missing. |
| Our | Crop prediction | Environmental and soil characteristics | Artificial neural networks | Accuracy more than 99% for both dataset. | More soil parameters can be considered such as nutrient retention capacity. |

### Funding

This work was carried out with the funding of "Cooperative Research Program for Agriculture Science and Technology Development (Project No. RS-2021-RD010360, Development of pests and plant diseases diagnosis using intelligent image recognition)" Rural Development Administration, Republic of Korea. The funders had no role in study design, data collection and analysis, decision to publish, or preparation of the manuscript.

### Grant Disclosures

The following grant information was disclosed by the authors:
"Cooperative Research Program for Agriculture Science and Technology Development (Project No. RS-2021-RD010360, Development of pests and plant diseases diagnosis using intelligent image recognition)" Rural Development Administration, Republic of Korea.

### Competing Interests

The authors declare there are no competing interests.

### Author Contributions

- Shabana Ramzan conceived and designed the experiments, performed the experiments, performed the computation work, prepared figures and/or tables, authored or reviewed drafts of the article, and approved the final draft.
- Basharat Ali conceived and designed the experiments, performed the experiments, performed the computation work, prepared figures and/or tables, and approved the final draft.
- Ali Raza conceived and designed the experiments, performed the experiments, performed the computation work, prepared figures and/or tables, authored or reviewed drafts of the article, and approved the final draft.
- Ibrar Hussain conceived and designed the experiments, performed the experiments, performed the computation work, prepared figures and/or tables, and approved the final draft.
- Norma Latif Fitriyani conceived and designed the experiments, analyzed the data, prepared figures and/or tables, authored or reviewed drafts of the article, and approved the final draft.
- Yeonghyeon Gu conceived and designed the experiments, analyzed the data, authored or reviewed drafts of the article, and approved the final draft.
- Muhammad Syafrudin conceived and designed the experiments, analyzed the data, prepared figures and/or tables, authored or reviewed drafts of the article, and approved the final draft.

### Data Availability

Data and code are available in the Supplemental Files.
The Crop Recommendation Dataset is available at Kaggle:
https://www.kaggle.com/datasets/atharvaingle/crop-recommendation-dataset.

## Supplemental Information

Supplemental information for this article can be found online at http://dx.doi.org/10.7717/peerj-cs.2478#supplemental-information.

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
