# Peer review of "An innovative artificial neural network model for smart crop prediction using sensory network based soil data"

_PeerJ Computer Science, doi:10.7717/peerj-cs.2478_

## Round 0.1 · original submission · Major Revisions

Introduction section should provide clear statements of the research goals. There and everywhere the authors should show what is novel in the approach and results.

Reviewer 1 ·

Basic reporting

The paper’s structure generally conforms to PeerJ standards, but there are areas that require improvement:

The introduction should provide more context and a clearer statement of the research gap being addressed.
Literature citations need to be more comprehensive, especially when introducing the research problem and methodology.

Experimental design

The proposed method lacks substantial innovation. The authors should clearly articulate what sets their approach apart from existing techniques in the field.
The description of Dataset 2 is insufficient. The authors need to provide detailed information about this dataset, including:
The types of data included
The number of data points
How the data was collected and preprocessed
Any relevant statistics or characteristics of the dataset?

Validity of the findings

The evaluation metrics seem appropriate, but there’s a lack of comprehensive comparison with state-of-the-art methods. A more detailed quantitative comparison would strengthen the paper.
The discussion of results is not sufficiently in-depth. The authors should explore the reasons behind their model's performance and discuss potential limitations.

Additional comments

The paper addresses a topic of practical importance, but the overall level of innovation is limited. The authors should emphasize the unique contributions of their work more clearly.
Some sections of the paper would benefit from clearer and more precise language. Consider having a native English speaker or a professional editing service review the manuscript.

The current quality of the paper is average and requires significant revisions before it can be considered for publication. The main issues are the lack of innovation in the proposed method, insufficient description of Dataset 2, and the need for a more in-depth analysis of the results. I suggest that the authors address these concerns thoroughly in a revised submission.

Reviewer 2 ·

Basic reporting

This paper introduces an ANN-based crop prediction system utilizing sensor-derived soil data, including nitrogen, phosphorus, potassium, temperature, humidity, pH, rainfall, electrical conductivity, and soil texture. Various machine learning models were assessed and compared with the proposed model, which demonstrated superior performance. Although the study is well-designed and clearly explained, several improvements can be made.

- In section 2 (Related Works), include statistics or data to emphasize the importance of crop prediction systems and the advantages of developing an accurate model. Support this section with recent references. Additionally, recommend summarizing the discussion in a table at the end, highlighting what is new in this study compared to existing literature on the same topic.

- For Figure 1, remove the white background and shadow. Consider increasing the quality and using black font color to enhance readability. For Figure 7, remove the triple dots "..." after AdaBoosting and update the figure accordingly, including font size and quality. Generally, increase the text size and quality of all figures.

-Consider presenting Figures 7 and 8 as tables for improved clarity.

-The proposed work and methodology are not well explained. Include a diagram illustrating how the Arduino platform connects with other sensors to provide readers with a clearer understanding.

-Include the versions of all libraries used in the study, such as Python, Keras, TensorFlow, etc. Additionally, recommend providing citations, references, or links for both datasets used in the study.

-In Algorithm 1 (page 9), replace the "0" in the left column with line numbers (1, 2, 3, etc.).

Experimental design

The experiment is clearly presented overall, but some improvements can be made. Include a diagram illustrating how the Arduino platform connects with other sensors to provide readers with a clearer understanding. Additionally, provide citations, references, or links for both datasets used in the study.

Validity of the findings

The proposed model was evaluated using two datasets, with results clearly presented and elaborated. Additionally, Table 7 offers a brief comparison of the CPS with existing works.

---

## Round 0.2 · accepted · Accept

The reviewers found your demonstrations and explanations satisfactory and recommended this version for publication in PeerJ Computer Science.

Reviewer 1 ·

Basic reporting

no comment

Experimental design

no comment

Validity of the findings

no comment

Additional comments

no comment

Reviewer 2 ·

Basic reporting

No comments

Experimental design

No comments

Validity of the findings

No comments